# Implementation of the Recovery Model and Its Outcomes in Patients with Severe Mental Disorder

**DOI:** 10.3390/healthcare12090952

**Published:** 2024-05-06

**Authors:** Antonio José Sánchez-Guarnido, María Isabel Ruiz-Granados, José Antonio Garrido-Cervera, Javier Herruzo, Carlos Herruzo

**Affiliations:** 1Mental Health, University Hospital Virgen de las Nieves, 18014 Granada, Spain; antonioj.sanchez.guarnido.sspa@juntadeandalucia.es; 2Department of Psychology, University of Cordoba, 14071 Cordoba, Spain; z22rugrm@uco.es (M.I.R.-G.); ed1hecaf@uco.es (J.H.); 3Mental Health, Hospital of Antequera, 29200 Málaga, Spain; josegarrido@uma.es

**Keywords:** mental health, recovery model, severe mental disorder, treatment, professional practices

## Abstract

Background: The recovery model assumes that the patient can experience personal growth even while maintaining symptoms of a mental disorder. In order to achieve this recovery, the practices of professionals must also change. However, in our setting, there are limited data on the implementation of practices based on the recovery model and their effect on personal recovery. Objective: To describe the association between professionals’ practices and patients’ personal recovery. Methods: An observational and cross-sectional study in which the Recovery Self-Assessment (RSA) was used to assess the degree of implementation of the different practices and the Recovery Assessment Scale (RAS) was used to assess the personal recovery of 307 patients with severe mental disorders. Results: Patients attended by professionals who followed the recovery model obtained a greater personal recovery (*p* < 0.001, d = 1.10). The dimension associated with greater recovery was that of working toward life goals. The least implemented dimensions had to do with offering treatment options and patient participation in decision-making. This study was conducted in accordance with STROBE (STrengthening the Reporting of OBservational studies in Epidemiology). Conclusions: Although this is a cross-sectional study that does not allow us to establish causal relationships, it shows that the model with which mental health professionals work is associated with patients’ chances of recovery. We therefore consider that it is important to foster the implementation of practices based on the recovery model within mental health care.

## 1. Introduction

The concept of recovery in mental health has evolved over time. In the past, recovery was understood as the absence of symptoms; however, this approach has been criticized for being too simple. An alternative perspective, termed personal recovery, sees recovery as a process carried out by a person with a serious mental illness to achieve psychological well-being beyond the limitations of the disease [1]. Recovery is understood from this perspective as a personal and ongoing process involving acceptance of the disability and the development of a new meaning and purpose in life [2,3,4].

The personal recovery model, often called simply the recovery model, is an important approach to mental health care because it recognizes that people living with mental disorders can achieve a full and satisfying life. Scientific evidence supports this approach by demonstrating that such a personal recovery is possible. Both in a 2011 study by Leamy et al. and in a subsequent review by the same authors [5], they found that people who had experienced personal recovery had a greater sense of identity, felt more confident in themselves and their abilities, and had a greater sense of direction and meaning in their lives. They also had more satisfying relationships with others and greater social participation, i.e., they were more involved in their community [5,6,7,8,9].

This scientific recognition has prompted health services to take an interest in implementing practices that can promote such recovery. Mental health services have traditionally been designed to reduce symptoms, but reducing symptomatology is different from supporting personal recovery. In fact, many people can experience personal recovery in the presence of ongoing symptoms. In this sense, different practices are needed for these new goals [10,11,12,13].

A recovery orientation of services refers to the extent to which services are designed to facilitate or promote patients’ personal recovery [14]. Attempts have been made to define what this service orientation means and to provide guidelines on how to promote this recovery [15]. The initial standards for recovery-oriented healthcare systems were published in the USA [16]. More recently, Davidson et al. developed a comprehensive and consistent set of standards or guidelines for recovery-oriented practice [14], as well as a measure of recovery orientation, the Recovery Self-Assessment [17].

Recovery-oriented practice has been implemented in different guides and manuals, such as “Personal Recovery and Mental Illness: A Guide for Mental Health Professionals” [18] or “Partnering for Recovery in Mental Health: A Practical Guide to Person-Centered Planning” [19]. In these guides, the main objectives for a recovery-oriented practice are defined as follows: (1) fostering relationships in patients that give them a sense of belonging and self-esteem; (2) conveying hope and providing models of recovery through contact with others who have achieved meaning in their lives; (3) focusing on strengths rather than deficits; (4) developing and maintaining a positive sense of self; (5) supporting patients in pursuing activities directed by their own values; and (6) empowering patients in self-care, advocacy, and social inclusion [19,20].

In general, the available evidence shows that personal recovery is influenced by various recovery-oriented practices, such as the defense of patients’ rights, especially in moments of greatest vulnerability, avoiding involuntary admission or treatment, physical restraints, or other violations of these rights. The recovery model also considers that services should offer a sufficient range of treatments to cover the different needs of each patient (psychological, social, occupational, etc.) instead of focusing on a single type of treatment, for example, pharmacological. It is also important that the patient can actively participate in making decisions about his or her treatment, including the treatment options available at the facility. It is important that the patient is also involved by jointly assessing with the professional the risks of each treatment option, including possible side effects, symptomatic worsening, or suicidal behavior. Regulated mutual support is also another aspect to be favored among practices based on this model. Finally, interventions from a recovery model should focus on getting the patient to find a new life project and supporting the patient in this direction with the resources he/she needs [14,17,18,21,22,23].

These advances have led to the study of how these organizational aspects and practices influence personal recovery. Among the countries where these practices have been most developed and studied are the USA, Canada, Australia, the United Kingdom, and Ireland. In these countries, specific programs have been developed, measurement instruments have been validated, and studies have been carried out to verify their efficacy [24]. In a recent systematic review where 309 studies on recovery practices were identified, these practices were associated with better results in personal recovery at the functional, existential, and social levels. Four common mechanisms were found in the practices that influence these outcomes: (1) providing information and skills; (2) promoting a working alliance; (3) role modeling recovery; and (4) increasing choice. This review concludes that recovery-oriented interventions propel patients toward personal recovery. On the contrary, when recovery practices are not applied, patients’ potential to develop a full life is limited [25]. Even so, the information available on the implementation of the recovery model in our context is still limited, and not enough is known about how these practices influence the personal recovery of people with mental disorders. More research is needed.

Therefore, the main objective of this study is to analyze the association between professional practices (practice behaviors) based on the recovery model and patients’ personal recovery. We also want to explore the degree of implementation of the different practices or factors (Life Goals, Involvement, Diversity of Treatment Options, Choice, and Individually Tailored Services) within the recovery model within the health system using the RSA-R questionnaire. Finally, the aim is to find out which dimensions of practice are most associated with recovery. Following the results obtained by Sklar et al. (2013) [26], it is hypothesized that patients treated by professionals who follow work guidelines within the recovery model will obtain better scores in personal recovery.

## 2. Methods

### 2.1. Design and Participants

An observational and cross-sectional study was carried out. This study was conducted in accordance with STROBE (STrengthening the Reporting of OBservational studies in Epidemiology).

The inclusion criteria were the following: Diagnosis of severe mental disorder (psychosis or bipolar disorder), age between 18 and 65 years, residence in Andalusia (Spain), and having basic proficiency in reading, writing, and comprehension of Spanish. The patients were in outpatient treatment within the Andalusian mental health network and were selected on the basis of a cluster sampling of the different units of the Andalusian mental health system.

### 2.2. Instruments

The *Recovery Assessment Scale (RAS)* [27] was originally developed by Corrigan and colleagues in the USA and published in 1999. It was created as a tool to measure personal recovery in people with severe mental illness. It is a self-report test composed of 41 items and divided into five factors: personal confidence and hope, willingness to ask for help, goal and success orientation, reliance on others, and no domination by symptoms. Each item is scored on a scale of 1 to 5, with 1 being “Strongly Disagree” and 5 being “Strongly Agree”. The total RAS score is calculated by summing the scores of all items. A higher score indicates a higher level of recovery. It has demonstrated good test-retest reliability (r = 0.88), good internal consistency (Cronbach’s alpha = 0.93), and convergent validity with measures of empowerment, self-esteem, social support, quality of life, and hope [27,28,29].

The *Recovery Self-Assessment Revised Version (RSA-R)* is a useful tool for mental health facilities wishing to assess their level of recovery orientation. It is an instrument developed by the Yale School of Medicine within the Yale Program for Recovery and Community Health. The questionnaire can be used to identify strengths and areas for improvement and to develop action plans to improve care. There are several versions for users, families, managers, and providers; in this research, a self-administered version for users is administered. The questionnaire consists of 32 items answered by the patient, divided into five factors: Life Goals, Involvement, Diversity of Treatment Options, Choice, and Individually Tailored Services. Each item is rated on a scale of 1 to 5, and a higher score on each factor means a higher implementation in the system of that group of practices. Cronbach’s alpha for the whole scale is 0.94. In our study, the first and fourth quartiles of the scores obtained in this questionnaire will be used to operationally define which patients are receiving a high or low recovery-oriented treatment [17].

### 2.3. Data Analysis

A descriptive analysis was carried out with the means of the different scores, and an analysis of comparison of means with Student’s *t*-test was performed, also finding Cohen’s d of the differences between groups. Cohen’s d value, also known as the effect size measure, is used to quantify the size of the difference between two groups on a variable. It is calculated by dividing the difference between the means of the two groups by the common standard deviation. This formula allows comparison of the effect size between different samples and studies. Cohen’s d values of 0.2, 0.5, and 0.8 have been established as conventions for classifying effect sizes as small, medium, and large [30]. SPSS 25 and a significance level of 0.01 were used for all analyses.

### 2.4. Ethical Considerations

Informed consent for participation in the study was obtained from all patients, all procedures adhered to the principles of the Declaration of Helsinki, and this study protocol was approved by the Clinical Research Ethics Committee of Andalusia (Spain).

## 3. Results

As shown in Table 1, a total of 307 people participated, of whom 186 were men and 121 were women. The mean age was 40.46 years (SD = 10.497), and the mean number of years of disease evolution was 15.6 (SD = 8.68). The distribution by diagnosis according to ICD-10 shows that 70% (*n* = 215) were diagnosed with psychotic disorders, 14% (*n* = 43) with personality disorders, and 11.1% with mood disorders.

The results shown in Table 2 show that patients treated according to the principles of the recovery model as assessed by the RSA-R obtain significantly higher overall personal recovery scores (*p* < 0.01) and also have a very high effect size (d = 1.10).

Analyzing the different dimensions of recovery practices, we see that they all have a statistically significant influence with moderate or high effects on personal recovery. The most influential practice in our sample is working on life goals (d = 1.12). The least influential is participation, but still with a moderate effect size (d = 0.69). All the others have high effects: treatment options (d = 0.86), rights and autonomy (d = 0.99), and individualization of treatment (d = 0.90).

Table 3 shows the mean scores on the level of use of the different practices by the professionals (1 to 5). The least utilized practices are related to participation (mean of 2.73 out of 5) and treatment options (mean of 3.00 out of 5). The other dimensions (life goals, rights and autonomy, and individualization of treatment) are above 4 out of 5.

## 4. Discussion

The main hypothesis of this study was the existence of a positive association between professional practices based on the recovery model and the personal recovery of people with severe mental disorders. The data obtained in this study support this hypothesis. Thus, the results show that patients who were cared for by professionals who followed practices based on the recovery model evaluated through the RSA-R obtained higher scores in personal recovery as measured by the RAS. This aligns with previous literature supporting these practices [5,26].

This study also assessed the association between personal recovery and different dimensions within professional practices. The results showed that recovery was greater when patients were offered a variety of treatment options, involved in making decisions about their treatment, focused on helping people achieve their life goals, respected patients’ rights and autonomy, and offered individualized treatment.

Of all the dimensions discussed, the factor with the greatest effect on recovery has been shown to be working toward life goals. This factor is understood as the degree to which the mental health facility focuses on helping people with mental disorders achieve their life goals. The goal of recovery is to live a full and meaningful life, and life goals are an important component of this. That is, people with mental disorders can be helped to identify their values, set realistic and achievable goals, develop a plan to reach those goals, and learn the skills necessary to reach them. Thus, for example, a mental health facility can offer vocational rehabilitation services and help patients develop the skills and confidence needed to return to work or school. These aspects are also treated with different psychotherapeutic models, such as acceptance and commitment therapy, which has already shown its effectiveness in the intervention of people with severe mental disorders [31].

Another important factor is respect for the rights of people with mental disorders. People with mental disorders have the same rights as any other person, and respect for their rights is fundamental to their recovery [32]. It is essential to avoid discrimination and stigmatization of people with mental disorders so that people can feel safe, avoiding coercive practices such as mechanical restraints.

Receiving individualized treatment is another important aspect of personal recovery. Each person with a mental illness is unique and has unique needs. Individualized treatment takes into account a person’s preferences, values, and goals, as well as his or her individual situation. Individualized treatment can help people overcome specific barriers to recovery. When people are having difficulty coping with a particular aspect of their mental illness, individualized treatment can help them develop the skills and resources needed to overcome that obstacle [20].

Although the three dimensions mentioned above—individualization of treatment, patients’ rights, and working with life goals—have shown a high degree of implementation in the Andalusian public health system, the present study has found that the practical implementation of the other two dimensions evaluated is much lower, namely: treatment options and patient participation in the choice of treatment.

In relation to the diversity of treatment options, this factor focuses on the range of treatment options available in a facility. The diversity of treatment options is important to ensure that people with mental disorders have access to the treatment that best suits their needs. Thus, for example, a mental health system that offers pharmacological, psychotherapeutic, social support, occupational, and community-based treatments may be more effective in helping people with mental disorders recover than a system that only offers one type of these treatments [8,33]. In addition, it is also important that the treatments offered are evidence-based so that people with mental disorders may have more options for finding a treatment that is effective for them personally [34].

Finally, the least implemented factor is patient participation. This factor refers to the degree to which people with mental disorders are involved in decision-making about their treatment. Patient involvement is important to ensure that people feel empowered and that their opinions are taken into account [8]. A mental health system could offer patients the opportunity to participate in the planning of their treatment as well as in the evaluation of their progress. In addition, it would also be interesting if they could participate in the decision-making regarding the activities offered in the range of services of the mental health system [35]. There is evidence that people who actively participate in making decisions about their treatment achieve greater commitment to it, a reduction in symptoms, better self-esteem, greater satisfaction, and lower hospitalization rates. Among the factors that previous research has observed that can reduce this participation is the lack of trust between the patient and the professional to be able to communicate their impressions about the treatment and between the professional and the patient to be able to listen to the patient’s vision of the treatment. Reducing these communication barriers and improving mutual trust within a therapeutic relationship could improve shared decision-making [23].

One limitation of this study is its cross-sectional design, which means that we can only observe associations, but we have no data to support a direction of causality. Even so, we consider that the contributions of the study may be relevant, given that it is the first study to evaluate the implementation of the recovery model in the Andalusian Health System. In addition, we also obtain data on how the different dimensions of professional practices are associated with personal recovery. The characteristics of the Andalusian Health System are unique, and therefore, we must be careful when generalizing certain conclusions, especially with respect to the degree of implementation of the recovery model in other health systems. We, therefore, recommend that similar studies be carried out in other health systems. For future research, it would be advisable to carry out longitudinal studies, both observational and experimental, that would allow us to contrast the causal relationships suggested in our study.

## 5. Conclusions

In conclusion, it can be affirmed that, within the public mental health system in Andalusia, the present study reinforces the idea that the model with which professionals work is important in terms of the results obtained with patients. It is, therefore, important to achieve widespread application of practices that are based on the philosophy of the recovery model. In this way, results can be achieved that go beyond symptom reduction, and people with severe mental disorders can lead meaningful lives.

## Figures and Tables

**Table 1 healthcare-12-00952-t001:** Description of the sample.

**Quantitative Variables**	**Mean**
Age (years)	40.46
Years of Disease Evolution	15.6
**Categorical Variables**	**Frequency**
**Sex**	
Men	186
Women	212
Total	307
**Diagnosis (ICD-10)**	**Frequency**
Psychotic Disorders (70%)	215
Personality Disorders (14%)	43
Mood Disorders (11.1%)	34

**Table 2 healthcare-12-00952-t002:** RAS (personal recovery) scores as a function of high or low scores on RSA dimensions (types of practices received).

RSA-R Dimensions	RAS in Patients with Low Scores on RSA-R Dimensions	RAS in Patients with High Scores on RSA-R Dimensions	Cohen’s *d* Value	*p*-Value
RSA total	148.929	180.690	1.103	*p* < 0.01
RSA life goals	146.938	179.253	1.122	*p* < 0.01
RSA participation	157.054	177	0.693	*p* < 0.01
RSA treatment options	152.6143	177.362	0.860	*p* < 0.01
RSA rights and autonomy	151.788	180.397	0.994	*p* < 0.01
RSA individualization of treatment	148.839	174.766	0.900	*p* < 0.01

**Table 3 healthcare-12-00952-t003:** Mean scores in RSA-R dimensions.

Dimensions	Mean	SD
Life goals	4.19	0.85
Participation	2.73	1.05
Treatment options	3.00	0.83
Rights and autonomy	4.19	1.04
Individualization of treatment	4.25	0.96

## Data Availability

The data presented in this study are available on request from the corresponding author. The data are not publicly available because they are part of an ongoing project.

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
