# Peer review of "Implementation of the Recovery Model and Its Outcomes in Patients with Severe Mental Disorder"

_healthcare, 2024, doi:10.3390/healthcare12090952_

Round 1

Reviewer 1 Report

Comments and Suggestions for Authors

Peer reviewed article:

Implementation of The Recovery Model and its Outcomes in Patients with Severe Mental Disorder

Thank you for the opportunity to review your insightful paper. Any work about Recovery Oriented Practice (ROP) and personal recovery is greatly needed in the limited literature internationally.

I thoroughly enjoyed reading your article. I could find no issues with the paper methodologically. It was thorough and the article was easily read.

There was not much about ROP explained in your paper. I would like you to elucidate on that a bit more. You touch on it but do not make it explicit enough. Your paper is clearly about ROP and personal recovery but it would be interesting to know the ROP models utilised by the practitioners implementing the care. There is plenty of literature on this. Perhaps you did not think some of it was relevant but I believe you could find some valuable insights into other work being done in this space.

Your hypothesis was good and it was supported. I agree ROP is successful and your paper goes a long way to demonstrate this, so I appreciate your work there.

You mentioned shared decision making in the introduction but this was nowhere in the methodology or discussion. It is an important point for ROP and further elucidation about this factor would be helpful. It is important that as far as care is given to people with mental health distress, that they have an opportunity to choose the treatments/medications they want or refuse. And whether they choose to take medications at all. In addition, shared decision making in risk-taking is very important for ROP.

I realise your paper was very well done in what it presented but a bit more breadth might have helped it with some extra findings about risk-taking and medication choice for service users.

You had a fairly good understanding on the ROP literature that is out there but if you look a bit further, there is more work on shared decision making and the importance of this in ROP and personal recovery.

There is a great deal of literature but here’s just a few.

For example:

Old but great is:

Deegan (2005). The importance of personal medicine: A qualitative study of resilience in people with psychiatric disabilities

Deegan (2006). Shared Decision Making and Medication Management in the Recovery Process

And more recent:

Ashoorian & Davidson (2021). Shared decision making for psychiatric medication management: a summary of its uptake, barriers and facilitators

Zisman-Ilani, Lysaker, & Hasson-Ohayon (2021). Shared risk taking: Shared decision making in serious mental illness

Again than you for the opportunity to review your paper. I believe just a tad more explanation of ROP and shared decision making as part of what you are trying to explain in ROP would be helpful. I realise it wasn’t directly related to your study but you mention it as important and you could explain how this part benefits in life goals and it would likn nicely in there. More ROP literature might enhance your paper also.

Author Response

Thank you for your comments. We agree with your suggestions.

We included further explanation of both Recovery Oriented Practice as well as shared decision making. We added information in both the introduction and discussion of the article.

We also added some references along these lines

Reviewer 2 Report

Comments and Suggestions for Authors

Dear authors,

Thank you for the manuscript on this important topic. With regard to the layout of the manuscript, it is noticeable that some questions remain unanswered. For example, how are the "different practices" (p. 2, line 84) operationalised? Unfortunately, the reader learns nothing about these approaches to practice. The statements made by the authors are not tenable in the present form. Furthermore, the results of the study are already reported in the methods section (p. 2, lines 94 ff.). Table 1, which usually contains the description of the sample, should be added. In addition, information on the setting in which the study was conducted is missing. Even in the discussion, the impression unfortunately persists that a group of patients completed two questionnaires, which were then compared with each other. The statement made in the discussion that the results show that patients "who were cared for by professionals who followed practices based on the recovery model" (p. 4, line 145 ff.) is not comprehensible on the basis of the results and it remains unclear exactly what this means. It also remains questionable what is meant, for example, by "different dimensions within professional practices" (p. 4, line 149 f.). These practices are not described anywhere in the manuscript. 

Overall, the reporting in the manuscript is not sufficiently comprehensible. Against this background, the manuscript as a whole should be comprehensively revised. The revision should be based on the valid reporting guidelines (https://www.equator-network.org/ ), in this case the STROBE criteria, using the corresponding checklist (https://www.equator-network.org/wp-content/uploads/2015/10/STROBE_checklist_v4_cross-sectional.pdf). 

Author Response

Thank you for your comments. We respond to the questions raised and comment below on the changes made to the document in relation to them.

  • Regarding the question on how "different practices" are operationalized, this is done through the Recovery Self-Assessment Revised Version (RSA-R) questionnaire and the five areas of the questionnaire (Life Goals, Involvement, Diversity of Treatment Options, Choice and Individually-Tailored Services) are assessed. We added this information on page 2, lines 84-88.
  • We added information about recovery practices in the introduction so that the reader can learn what this approach consists of. We tailored the amount of information added to the brief report format chosen and the objectives of the manuscript.
  • We added a summary table of descriptive sample data.
  • We added information about the context in which the study is conducted (First method paragraph).
  • In the first paragraph of the discussion we add that the retrieval practices refer to those assessed by the RSA-R questionnaire. More detailed information about them appears in the description of the instrument and in analysis later in the discussion starting in the third paragraph.
  • Following their recommendation, we attach the STROBE checklist.

Reviewer 3 Report

Comments and Suggestions for Authors

General comment

 The authors attempt to validate the personal recovery model which is important given that a complete absence of mental health symptoms is unobtainable for some, and many can still live a fulfilling life growing despite these issues. Through an observation, cross-sectional study, the authors are attempting to describe a possible relationship between the various recovery models that lead to improved outcomes. However, the abstract should clearly state that the main limitation here is that this is a cross-sectional study that cannot demonstrate causality for clarity purposes. The authors should describe the objective of the study in more detail and the rationale and relevance for using the selected scales since these scales are not clinically utilized.

Specific Comments

·         Abstract

1.      Consider rewording objective to: “To describe the association between professionals’ practices and patients’ personal recovery.”

2.      Consider adding a statement following the end of Line 21 with some of the major limitations of the study design-wise.

·         Article text

1.      Introduction:

a.      Page 2, Line 75: Consider providing clarification to how research has been done internationally in other countries and how this may apply outside of Andalusia

b.      Page 2, Line 68: Consider adding in a description of when the personal recovery model falters and is less accurate or falters.

c.       Page 2, Lines 82-83: Please define “professional practices” – a reader could interpret this as individuals, practice behaviors or the facilities themselves.

2.      Methods:

a.      Page 5, Lines 100-113: It is unclear how the RAS and RSA-R were selected for use in this study. Some context of these scales and the populations they were studied would provide some relevance to the current study.

b.      Page 5, Lines 106-107: It is also unclear who is completing the two scales. The authors state that RSA-R is “a useful tool for mental health facilities” – please clarify whether these are administrators, providers/professionals or patients.

c.       Consider adding an explanation on the historical context or reasoning being the selection for these d value cutoffs. For the average reader, it may not be clear what the association between d value and effect as compared to other studies.

d.      Page 3, Line 100: Define scoring and how to interpret scores based on quantitative measurement historically.

e.      Page 3, Line 106: Define scoring and how to interpret scores based on quantitative measurement historically.

3.      Discussion:

a.      Page 5, Line 181: Consider mentioning possible broad, cultural differences in the Andalusian public health system and how this may affect the external validity of the study to other healthcare systems internationally.

b.      Page 5, Line 209: Consider breaking up the statement to increase conciseness for the concluding statement.

4.      Consider adding a conclusion section with the last paragraph separate from the discussion section and review for grammatical errors that prohibit understanding.

5.      Table 2: Please provide a footnote to define the scoring of the Dimensions so that the reader knows how to interpret the mean scores.

Comments on the Quality of English Language

The manuscript is grammatically written correctly without significant typographical errors. However, phrasing of various statements could be improved for clarity.

Author Response

General comment

 The authors attempt to validate the personal recovery model which is important given that a complete absence of mental health symptoms is unobtainable for some, and many can still live a fulfilling life growing despite these issues. Through an observation, cross-sectional study, the authors are attempting to describe a possible relationship between the various recovery models that lead to improved outcomes. However, the abstract should clearly state that the main limitation here is that this is a cross-sectional study that cannot demonstrate causality for clarity purposes. The authors should describe the objective of the study in more detail and the rationale and relevance for using the selected scales since these scales are not clinically utilized.

R: Thank you for your comments, we find them very useful and have taken them into account. We added the limitation in the summary, changed the target description and expanded information on the scales. 

Specific Comments

  • Abstract
  1. Consider rewording objective to: “To describe the association between professionals’ practices and patients’ personal recovery.”

R: Thank you for your comment. We have changed it.

  1. Consider adding a statement following the end of Line 21 with some of the major limitations of the study design-wise.

R: Thank you for your comment. We add as a main limitation the difficulty in establishing causal relationships given that this is a cross-sectional study.

Article text

  1. Introduction:
  2. Page 2, Line 75: Consider providing clarification to how research has been done internationally in other countries and how this may apply outside of Andalusia

R: Thank you for your comment. We add information on the application of the model in other countries and international research.

  1. Page 2, Line 68: Consider adding in a description of when the personal recovery model falters and is less accurate or falters.

R: Thank you for your comment. We added information on the consequences of when the application of the model fails.

  1. Page 2, Lines 82-83: Please define “professional practices” – a reader could interpret this as individuals, practice behaviors or the facilities themselves.

R: Thank you for your comment. We refer to "practice behaviors" we added it in parentheses in the text for clarification.

  1. Methods:
  2. Page 5, Lines 100-113: It is unclear how the RAS and RSA-R were selected for use in this study. Some context of these scales and the populations they were studied would provide some relevance to the current study.

R: Thank you for your comment. Information on the context of these scales has been added.

  1. Page 5, Lines 106-107: It is also unclear who is completing the two scales. The authors state that RSA-R is “a useful tool for mental health facilities” – please clarify whether these are administrators, providers/professionals or patients.

R: Thank you for your comment. We clarify in the text that both scales are completed by the patients.

  1. Consider adding an explanation on the historical context or reasoning being the selection for these d value cutoffs. For the average reader, it may not be clear what the association between d value and effect as compared to other studies.

R: Thank you for your comment. We have added an explanation.

  1. Page 3, Line 100: Define scoring and how to interpret scores based on quantitative measurement historically.

R: Thank you for your comment. We have added information on punctuation and its interpretation.

  1. Page 3, Line 106: Define scoring and how to interpret scores based on quantitative measurement historically.

R: Thank you for your comment. We have added information on punctuation and its interpretation.

  1. Discussion:
  2. Page 5, Line 181: Consider mentioning possible broad, cultural differences in the Andalusian public health system and how this may affect the external validity of the study to other healthcare systems internationally.

R: Thank you for your comment. We have added comments on this in the limitations section of the study and in recommendations for future studies.

  1. Page 5, Line 209: Consider breaking up the statement to increase conciseness for the concluding statement.

R: Thank you for your comment. We have divided it.

  1. Consider adding a conclusion section with the last paragraph separate from the discussion section and review for grammatical errors that prohibit understanding.

R: Thank you for your comment. We separate the conclusion in a separate section.

  1. Table 2: Please provide a footnote to define the scoring of the Dimensions so that the reader knows how to interpret the mean scores.

R: Thank you for your comment. The final version of the manuscript has been revised by a native English-speaking translator.

Round 2

Reviewer 1 Report

Comments and Suggestions for Authors

Hi

Thank you for the opportunity to go through this process with you. I still think you could elevate recovery-oriented practice (ROP) and shared decision making (SDM) due to these being highly necessary in light of personal recovery and outcomes for people living with mental distress. However, I know words are tight and you have improved this to an appropriate level. 

I agree in this paper being published if the editor sees fit to do so as it is a highly important issue that is requiring much more focus on it in building a research base for ROP and SDM.

Kind regards

Janice

Reviewer 2 Report

Comments and Suggestions for Authors

Dear authors,

Thank you very much for the revised manuscript. The revisions are comprehensive and comprehensible. The additions to the introduction are sensible but also very extensive, which makes the introduction very long overall and in comparison to the other sections. The presentation of the results is rather brief overall. In the "Methods" section on p. 3 (lines 124-129), the results of the study are already reported ("A total of 307 people participated..."). These should be moved to the results section. Furthermore, Table 1 on p. 3 should be integrated into the text or reference should be made to the table in the text. The manuscript should refer to the use of the Reporting Guideline (https://www.equator-network.org/reporting-guidelines/strobe/ ). With regard to the handling of sources, the first name on p. 3 (line 132) can be deleted. 

Author Response

Thank you very much for the revised manuscript. The revisions are comprehensive and comprehensible. The additions to the introduction are sensible but also very extensive, which makes the introduction very long overall and in comparison to the other sections

Thank you for your comments. We have slightly reduced the addictions made in the fifth paragraph of the introduction.

In the "Methods" section on p. 3 (lines 124-129), the results of the study are already reported ("A total of 307 people participated..."). These should be moved to the results section.

Thank you for your comments. We have moved it to the results section.

Furthermore, Table 1 on p. 3 should be integrated into the text or reference should be made to the table in the text.

Thank you for your comments. We have included the reference to the table in the text.

The manuscript should refer to the use of the Reporting Guideline (https://www.equator-network.org/reporting-guidelines/strobe/ ).

Thank you for your comments. We have referred to the use of the Reporting Guideline in the summary and in the method section.

With regard to the handling of sources, the first name on p. 3 (line 132) can be deleted.

Thank you for your comments. We have deleted the first name in line 132